**XBAER derived aerosol optical thickness from OLCI/Sentinel-3 observation**

Linlu Mei[1], Vladimir Rozanov[1], Marco Vountas[1], John P. Burrows[1], Andreas Richter[1]

[1]Institute of Environmental Physics, University of Bremen, Germany

**Abstract**

A prolonged pollution haze event occurred in the northeast part of China during December 16 - 21, 2016. To assess the impact of such events, the amounts and distribution of aerosol particles formed in such events need to be quantified. The newly launched Ocean Land Color Instrument (OLCI) onboard Sentinel-3 is the successor of the MEdium Resolution Imaging Spectrometer (MERIS). It provides measurements of the radiance and reflectance at the top of the atmosphere which can be used to retrieve the Aerosol Optical Thickness (AOT) on both synoptic to global scales. In this paper, the recently developed AOT retrieval algorithm - eXtensible Bremen AErosol Retrieval (XBAER) has been applied to data from the OLCI instrument for the first time to inlustrate the feasibility of transferring XBAER to new instrument. The first global retrieval results show similar patterns as MODIS and MISR aerosol products. The AOT retrieved from OLCI is validated by comparison with AERONET observations and a correlation coefficient of 0.819 and bias (root mean square) of 0.115 is obtained. The haze episode is well-captured by the OLCI-derived AOT product. XBAER is shown to retrieve AOT from the observations of MERIS and OLCI.

88

## 1 Introduction

Haze is an atmospheric phenomenon which is associated with horizontal visibilities of less than l0 km and atmospheric relative humidity (RH) less than 90 % (Liu et al., 2013). Haze occurs as a result of pollution i.e. the release of sulfur dioxide ($SO_2$), nitrogen oxides ($NO_x$) and particles or the photochemical production of atmospheric particles (Sezer et al., 2005; Pudasainee et al., 2006). These particles are called aerosol. Aerosol has a variety of effects on climate and environment both directly and indirectly. The direct effect is through scattering which cools the atmosphere and surface system or by absorption of incoming solar radiation which also cools the surface but warms the atmosphere. Indirectly, aerosol impacts on cloud formation and the microphysical properties of clouds, which in turn influence cloud albedo and precipitation (Li et al., 2011) adding to therir negative health impacts. Aerosols are also the carriers of toxic substances such as heavy metals and polycyclic aromatic hydrocarbons (Wilkomirski et al., 2011). In Beijing, under high pollution conditions, the concentrations of sulfate and nitrate have been shown to account for 1/3 of the particle matter ($PM_{10}$) mass and 2/3 of the $PM_{2.5}$ mass, a part of which is attributed to the additional secondary conversion of $SO_4^{2-}$ from $SO_2$ and $NO_3^-$ from $NO_x$ (Ji et al., 2012). Haze has a significant effect on regional climatic phenomena, such as monsoon (Chung et al., 2002; Evan, et al., 2011) and on the environment e.g. air quality (Lin et al., 2012) and visibility (Zhao et al., 2011). Aerosol can adversely affects human health (Evan, et al., 2011), especially for the elderly, children (American Academy of Pediatrics Committee on Environmental Health, 1993), and even the new-born children (Dadvand et al., 2013).

A thick smoke haze enveloped the Eastern and Northern part of China in December 2016. Pictures taken by cameras onboard the satellite TERRA/AQUA show that the area affected by

haze exceeded about 1.5 million square kilometers area of China. The poor visibility resulted in
several highways and regional airports being closed for extended periods. The situation
deteriorated significantly during the haze event and became a matter of public concern.
Satellite observations of the reflectance of solar radiation at the top of the atmosphere are
used to determine Aerosol Optical Thickness (AOT), which is used as an indicator of air quality
(Kaufman et al., 2002). There are numerous attempts for the retrieval of aerosol properties from
satellite observations. AOT retrieval algorithms have been developed for use with the
measurements of Moderate Resolution Imaging Spectroradiometer (MODIS) (e.g. Dark-Target
(Levy et al., 2013), DeepBlue (Hsu et al., 2013), the Multiangle implementation of atmospheric
correction (MAIAC) (Lyapustin et al., 2011)),  Advanced Along-Track Scanning Radiometer
(AATSR) (e.g. AATSR Dual-Viewing (ADV) (Kolmonen et al., 2016; Sogacheva et al., 2017),
Oxford-RAL Aerosol and Cloud (ORAC) (Thomas et al., 2009) and Swansea University (SU)
(North et al., 1999) algorithms). AOT is also derived from observations of the Multi-angle
Imaging SpectroRadiometer (MISR) (Diner et al., 2005), PARASOL's Polarization and
Directionality of the Earth's Reflectances (POLDER) (Dubovik et al., 2014), Sea-Viewing Wide
Field-of-View Sensor (SeaWiFS) (Sayer et al., 2012) etc.
One challenge for the derivation of AOT long term datasets from satellite observation is to
generate comparable AOT data products from the different instruments, which have limited
lifetimes. Consequently mature aerosol algorithms, which can be applied to data from instruments
on different platforms, are required. For example, the three MODIS aerosol algorithms have been
applied to the Visible Infrared Imaging Radiometer Suite (VIIRS) instrument and the three

AATSR algorithms have been proposed to be applied to the observations of the Sea and Land
Surface Temperature Radiometer (SLSTR) instrument (Popp et al., 2016).

The MERIS instrument onboard Environmental Satellite (Envisat) provided valuable

information for different applications (Verstraete et al., 2010). There are several previous attempts
to develop AOT retrieval algorithms for MERIS, e.g. the Bremen AErosol Retrieval (BAER; von
Hoyningen-Huene et al., 2003, 2011), and the European Space Agency (ESA) standard aerosol
retrieval (Santer et al, 2007). These had mixed success (Mei et al., 2016a). BAER has limited
accuracy away from dark-vegetated surfaces and primarily for non-absorbing aerosols (de Leeuw
et al., 2015; Holzer-Popp et al., 2013) while the ESA standard AOT retrieval tends to overestimate
AOT (de Leeuw et al., 2015). The recently developed eXtensible Bremen AErosol (XBAER)
algorithm (Mei et al., 2016a , 2016b) has been internally validated in the Aerosol- Climate Change
Initiative (Aerosol-CCI) project (Popp et al., 2016), and shows very promising results.

The newly launched (on 16$^{th}$ Feb, 2016) instrument Ocean Land Color Instrument (OLCI)

takes the heritage of MERIS as it contains all MERIS channels. Theoretically it is possible to
transfer the mature MERIS retrieval algorithms to the OLCI instrument. In this paper, the XBAER
algorithm has been applied to OLCI instrument for the first time. To our best knowledge, this is
the first publication of AOT retrieved from OLCI. Although Sentinel-3 has only recently been
launched, applying XBAER to OLCI data we have identified a haze event over Beijing, China
during December 2016. We use observations by OLCI during this episode to test our retrieval of
AOT.

In this manuscript, the characteristics of OLCI and MERIS instruments are presented and

compared in Section 2. The XBAER algorithm is briefly explained in Section 3. Section 4 shows
the comparison between OLCI and MERIS instruments, first XBAER OLCI-derived AOT results
and a comparison with AOT from MODIS/MISR and AERONET observations is shown and
discussed from a global point of view. The AOT retrieved during the regional haze event is also
presented and discussed in Section 4. The conclusions are given in Section 5.
**181 2 OLCI instrument**

The European Space Agency Sentinel-3 satellite was successfully launched on $16^{th}$ February 2016.
It is one element of the EU Copernicus system previously known as the Global Monitoring for
Environment and Security (GMES) system
(https://sentinel.esa.int/web/sentinel/user-guides/sentinel-3-olci). The aim of the Sentinel-3
mission is to provide data continuity of observation and data products for two of the instruments
aboard ENVISAT viz MERIS
(https://earth.esa.int/web/guest/missions/esa-operational-eo-missions/envisat/instruments/meris)
and AATSR
(https://earth.esa.int/web/guest/missions/esa-operational-eo-missions/envisat/instruments/aatsr).
There is no overlap of observations because ENVISAT was lost unexpectedly and suddenly in
April, 2012. The outstanding performance of ENVISAT over the last decade led both scientists
and engineers to believe that it is valuable to make use of multiple sensing instruments to
accomplish its operational mission for oceanography & global land applications. The instruments
onboard Sentinel-3 include SLSTR (Sea and Land Surface Temperature Radiometer), OLCI
(Ocean and Land Colour Instrument), SRAL (SAR Altimeter), DORIS (Doppler Orbitography and
Radiopositioning Integrated by Satellite), and MWR (Microwave Radiometer), which can deliver
additional information for Sea/Land colour data (at least MERIS quality), Sea/Land surface
temperature (at least AATSR quality) , Sea surface topography data (at least Envisat RA quality)
(https://earth.esa.int/web/guest/missions/esa-eo-missions/sentinel-3).
The primary objective of OLCI is to observe the ocean and land surface in the solar spectral
region and thereby to harvest information related to biology. OLCI also provides information on
the atmosphere and contributes to climate studies. OLCI is a push-broom imaging spectrometer
that measures solar radiation reflected by the Earth, at a ground spatial resolution of 300 meter, in
21 spectral bands between 0.4 and 1.02 μm, with a swath width of 1270 km. A comparison
between the MERIS and OLCI instruments has been included in Table 1.
**Table 1 Spectral channels for MERIS and OLCI instruments**

| OLCI | | | MERIS | | | Usage |
|---|---|---|---|---|---|---|
| Band # | Central wavelength | Width | Band # | Central wavelength | Width | Δ: cloud * Surface ¶:Aerosol |
| 1 | 400 | 15 | | | | |
| 2 | 412.5 | 10 | 1 | 412.5 | 10 | Δ/*/¶ |
| 3 | 442.5 | 10 | 2 | 442.5 | 10 | */¶ |
| 4 | 490 | 10 | 3 | 490 | 10 | */¶ |
| 5 | 510 | 10 | 4 | 510 | 10 | */¶ |
| 6 | 560 | 10 | 5 | 560 | 10 | */¶ |
| 7 | 620 | 10 | 6 | 620 | 10 | */¶ |
| 8 | 665 | 10 | 7 | 665 | 10 | */¶ |
| 9 | 673.75 | 7.5 | | | | |
| 10 | 681.25 | 7.5 | 8 | 681.25 | 7.5 | */¶ |
| 11 | 708.75 | 10 | 9 | 708.75 | 10 | */¶ |
| 12 | 753.75 | 7.5 | 10 | 753.75 | 7.5 | Δ |
| 13 | 761.25 | 2.5 | 11 | 760.625 | 3.75 | Δ |
| 14 | 764.375 | 3.75 | | | | |
| 15 | 767.5 | 2.5 | | | | |
| 16 | 778.75 | 15 | 12 | 778.75 | 15 | |
| 17 | 865 | 20 | 13 | 865 | 20 | |
| 18 | 885 | 10 | 14 | 885 | 10 | * |
| 19 | 900 | 10 | 15 | 900 | 10 | |
| 20 | 940 | 20 | | | | |
| 21 | 1020 | 40 | | | | |



## 3 XBAER algorithm

The XBAER algorithm was designed for the retrieval of AOT from MERIS and similar observations. It has its own cloud screening approach, aerosol type selection and surface parameterization (Mei et al., 2016a, 2016b). The cloud screening algorithm minimizes cloud contamination for aerosol retrieval in XBAER. The XBAER cloud masking algorithm determines the presence of cloud by using i) the brightness of the scene, ii) the homogeneity or variability of the top of the atmosphere reflectance and iii) cloud height information (Mei et al., 2016b). The threshold values in the XBAER cloud masking algorithm are selected by a two steps process. The ranges for the thresholds were determined by using accurate radiative transfer modeling with different surface and atmospheric scenarios. A histogram analysis has been used for different cloud, aerosol and surface scenarios to estimate the optimal threshold values for each criterion.

The XBAER algorithm uses a generic one-parametric surface parameterization for both land and ocean. XBAER uses a set of space-time dependent spectral coefficients to describe surface properties. The spatial and temporal resolutions are 10 km and monthly, respectively. The surface spectral reflectance can be determined simultaneously with AOT in an iterative procedure (Mei et al., 2016a). This approach assumes that the wavelength-dependent properties of surface spectral reflectance are constrained by space and time dependent spectral coefficients. The wavelength-independent single parameters (Soil-adjusted Vegetation Index (SAVI) for land retrieval and Normalized Differential Pigment Index (NDPI) for ocean retrieval) have been used as the "tuning" parameters. In this manner, XBAER is not limited to dark surfaces (ocean, vegetation) and also retrieves AOT over bright surfaces (e.g. desert, semiarid, and urban areas).

XBAER uses MODIS Dark-Target aerosol type assumptions and the expected aerosol type

for a given region and season is taken from an analysis of Aerosol Robotic Network (AERONET)
and Maritime Aerosol Network (MAN) observations for both land and ocean. AOT and surface
reflectance are retrieved by minimizing the difference between simulated and measured TOA
reflectance using a Look-Up-Table (LUT), created by the radiative transfer software package
SCIATRAN (Rozanov et al., 2014). Details of the XBAER algorithm can be found in Mei et al.,
(2016a, 2016b). A post-processing technique used in Aerosol-CCI project and the MODIS
monthly snow fraction dataset have been additionally applied to avoid unresolved clouds/snow
(Popp et al., 2016).
**4 Results**
**4.1 Verification**
One important characteristic invetsigated is the instrument spectral response function (SRF)
because it is the major difference between MERIS and OLCI for overlap channels. Fig. 1 shows
the SRF for the MERIS and OLCI overlap channels. The OLCI RSF mean dataset
([https://sentinel.esa.int/web/sentinel/technical-guides/sentinel-3-olci/olci-instrument/spectral-respo](https://sentinel.esa.int/web/sentinel/technical-guides/sentinel-3-olci/olci-instrument/spectral-respo)
[nse-function-data](https://sentinel.esa.int/web/sentinel/technical-guides/sentinel-3-olci/olci-instrument/spectral-response-function-data)) has been used. Differences between MERIS and OLCI SRF are identified but
have negligible impact on the retrieved AOT.

In order to quantitatively investigate the impact of different SRFs, the TOA reflectances have

been simulated with and without taking SRF into account. The simulations have been determined
by undertaking radiative transfer simulations using SCIATRAN for atmospheric and surface
conditions (Rozanov et al., 2014).The MERIS observation geometry for the 2nd July 2009 over
Paris was used to perform a forward simulation. In particular, the solar zenith angle, viewing angle
and relative azimuth were set to (32.32°, 28.7°, 30.65°) as suggested in Mei et al. (2016a).

In order to design representative simulated scenarios, we define a comprehensive set of

aerosol optical parameters, surface spectral reflectances, and other atmospheric properties
comprising temperature and pressure profiles, the profiles of the concentration of gaseous
absorbers and scattering.  Suitable ranges of values for all relevant inputs for the RTM are
obtained by statistical analysis of corresponding global products (Mei et al., 2016a). For this
purpose, we use:

**Surface reflectance**: Three typical surface types representing vegetation, soil and water, i.e.

relatively dark land (vegetation-covered city), bright land (desert), and water surface (ocean
surface), were used. The typical vegetation and soil spectra are adapted from von
Hoyningen-Huene et al. (2011), the liquid water spectrum comes from the SCIATRAN database
(see references in Rozanov et al., 2014). Fig. 2 shows the corresponding surface reflectance
spectra for selected surface types.

**Aerosol Scenarios**: Within the ESA Aerosol-CCI project, a representative value for global

mean AOT of 0.25 has been selected (Holzer-Popp et al., 2013; de Leeuw et al., 2015). Thus an
AOT of 0.25 was selected for the simulation of "vegetation" and "water" cases. An AOT value of
0.5 was used for the "soil" scenario to represent a 'real' case for the Sahara region. Moderately
absorbing (fine mode radius $r_{v,f}$ = 0.150 μm, coarse mode radius $r_{v,c}$ = 3.19 μm, fine mode variance
$\sigma_f$ = 0.408, coarse mode variance $\sigma_c$ = 0.754, fine/coarse mode volumes ($\mu m^3/\mu m^3$) are 0.055 and
0.038), pure maritime type ($r_{v,f}$ = 0.150 μm, $r_{v,c}$ = 3.19 μm, $\sigma_f$ = 0.408, $\sigma_c$ = 0.754, fine/coarse
mode volumes ($\mu m^3/\mu m^3$) are 0.04 and 0.296) and dust aerosol model ($r_{v,f}$ = 0.140 μm, $r_{v,c}$ = 1.74
μm, $\sigma_f$ = 0.454, $\sigma_c$ = 0.687, fine/coarse mode volumes ($\mu m^3/\mu m^3$) are 0.02 and 0.157) were used
for aerosol types.
**Other atmospheric parameters**: The profiles of temperature, pressure, and concentration of
the gases ozone, $O_3$, nitrogen dioxide, $NO_2$, and molecular oxygen, $O_2$ and water vapor, $H_2O$,
which all absorb in the 400 – 900 nm spectral region were provided by the Bremen 2D chemical
transport model (Sinnhuber et al., 2009).
In Table 1 the spectral channels of OLCI and MERIS are given. Fig. 2 (a) presents the
surface spectral reflectance for the three surface types selected. Fig.2 (b) presents the simulated
TOA differences for the above scenarios. The differences for all surface/atmospheric conditions
are less than 1.5%. These are similar to the simulation with and without convolution for MERIS
with the exception of the O2A and water vapor channels. However, the potential impacts of
different SRFs may also introduce some uncertainties to the XBAER cloud mask due to relative
strong impact of SRF to the O2A channels (about 20% difference).

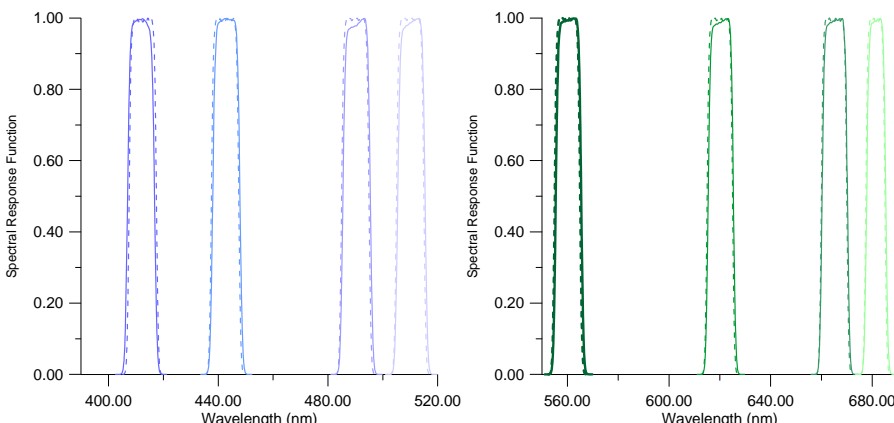


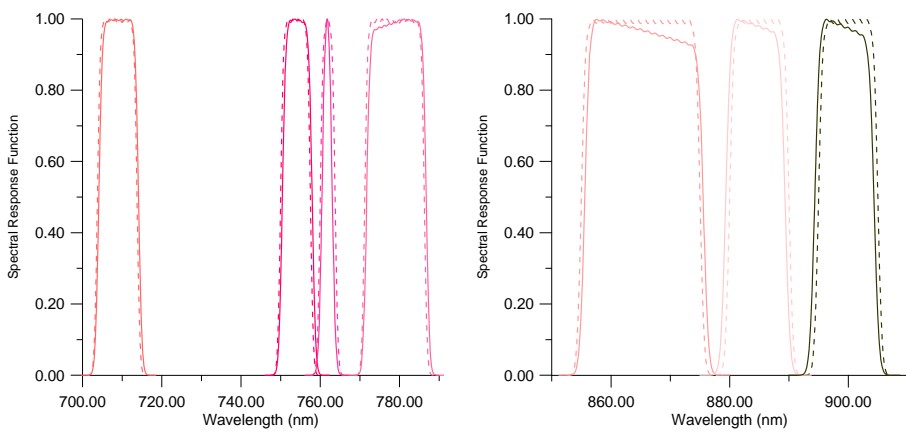

Fig 1 Spectral Response Function of MERIS (dash lines) and OLCI (solid lines) for overlap
channels

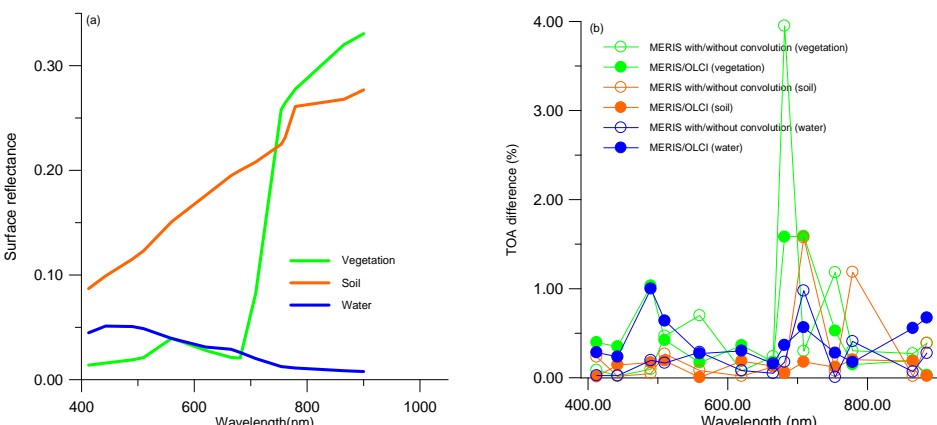

Fig 2 (a) Surface reflectances of the three selected surface types; (b) Comparisons of the simulated
TOA reflectance for different combinations of MERIS and OLCI SRF values. Green, yellow and
blue colors in (a) and (b) represent vegetation, soil and water simulations. Filled circles in (b) are
differences for simulations using MERIS and OLCI SRFs. Circles in (b) are differences for
simulations with and without convolution.

## 4.2 First XBAER AOT retrieval for OLCI and its validation


AERONET observations are considered to be the "ground truth" for satellite validation (Holben et
al., 1998). Here, we collocate the XBAER OLCI aerosol retrievals with the AERONET Version
3.0 (https://aeronet.gsfc.nasa.gov/new_web/Documents/AERONET-V3_News_Final.pdf. last
access: 15 May, 2017), Level 1.5 (Level 2.0 for both AERONET Version 2.0 and 3.0 are not
available till 15 May, 2017) (Holben et al., 1998; Smirnov et al., 2000). As AERONET does not
provide AOT at 0.55 μm, data are interpolated to 0.55 μm using quadratic fits on a log-log scale
(Eck et al., 1999). Since AERONET provides a point measurement with high-temporal resolution
while satellite observations represent a 'regional' measurement depending on the satellite spatial
resolution for a particular overpass time, spatial statistics for the OLCI data are calculated and
compared to the temporal statistics of the AERONET observations taken within ±30 min of OLCI
overpass following the spatial-temporal technique of Ichoku et al. (2002).

Fig. 3 is a plot which compares XBAER-derived and AERONET observed AOT at 0.55 μm.

The collocations of Fig.3 contain various surface and aerosol types, which ensure a wide
representativeness of the validation. 733 collocations were found for December 2016. The colour
of each ordered pair ($0.025 \times 0.025$ increment) represents the number of such matchups. Negative
AOT (> -0.1) values are possible and reasonable as a result of the noise in satellite observations
(Levy et al., 2007) and uncertainties of surface parameterization. The comparison here excluded
negative values and only AOT values between 0.0 and 2.5 are used following the validation
method of other aerosol products (Sayer et al., 2012; Levy et al., 2013). The validation contains
various surface and aerosol types, which ensures a wide representativeness of the validation. The
regression equation is $y = (0.81x \pm 0.04) + (0.11 \pm 0.01)$ with slightly higher correlation compared
to the first MERIS validation ($R$=0.82 vs R=0.78) (Mei et al., 2016a).   The AOT is
reasonable-correlated between the two datasets (R = 0.82), with increased scatter for high aerosol
loadings. The majority of the data (87.5%) are for low aerosol loadings (AOT<0.3). The
comparison between XBAER AOTs and AERONET observations shows the acceptable quality of
the first OLCI XBAER results.

Fig. 4 shows the global monthly AOT of December 2016 for MODIS collection 6 (Levy et al.,

2013), MISR (Diner et al., 2005) and OLCI (XBAER) algorithm. In order to identify biomass
burning events, the active fire points of MODIS
(https://lance.modaps.eosdis.nasa.gov/cgi-bin/imagery/firemaps.cgi) are added to the figures.
Please note that the MISR "FIRSTLOOK" product is used because the monthly Land Surface and
Aerosol products are not yet processed for December 2016 (Personal communication with NASA
Langley ASDC on 16 February 2017). MODIS/MISR on board of TERRA and OLCI on board of
Sentinel-3 have very similar overpass time (within 30 minutes difference). Therefore, all four
results should show similar patterns for large AOT from desert dust events over Sahara, biomass
burning over West Africa and Amazon region and anthropogenic pollution over India and East
Asia. In Fig.4, XBAER AOT from OLCI shows similar patterns as the AOT from MODIS and
MISR for both land and ocean. However, there are differences in the magnitude of the AOTs.
Biomass burning over Africa, as observed in the MODIS active fire product, produces a 'plume
belt' of high AOT near the equator. This is observed in all three AOT products. The AOT
distribution pattern over India, which depends on the unique meteorological conditions and
emissions is captured by the three AOT data products as well. MODIS and OLCI show similar
pattern and magnitude of large AOT over Eastern China while the values from MISR are slightly
lower, which may be due to the relative small sampling compared to MODIS and OLCI. However,
the retrievals from XBAER over Australia are higher than those of MODIS and MISR. In addition
to potential contamination by thin clouds observed in the RGB composite figures, the calibration
uncertainties associated with a new instrument may also contribute to the bias of XBAER derived
AOT. The large AOT differences over Sahara may go, in part, back to different assumptions in the
different algorithms for bright surfaces (Lyapustin et al., 2011b; Mei et al., 2016a). Different
patterns over Amazon can most likely be attributed to the use of different cloud screening methods.
The global patterns obtained indicate that the generic XBAER algorithm works over both dark and
bright surfaces using its flexible surface parameterization approach. For relative dark surfaces, the
one-parametric surface parameterization is dominated by the first term (SAVI or NDPI tuned term)
making XBAER behave like the Dark-Target-like retrieval algorithm. For bright surfaces such as
desert, XBAER becomes similar to the DeepBlue AOT retrieval algorithm.

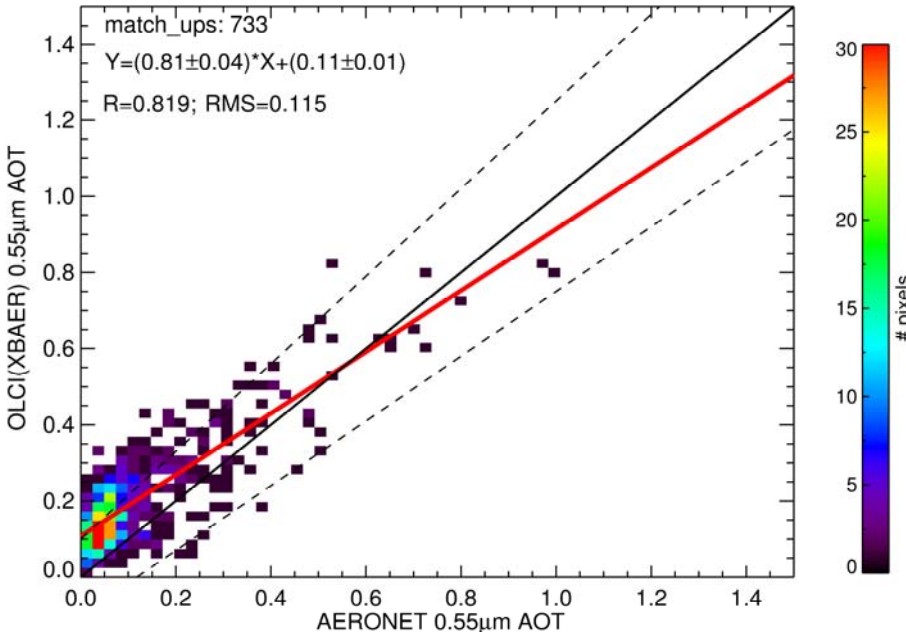



Fig.3 Global comparison of OLCI XBAER AOT with AERONET observations for 2016
December. R and 'match_ups' refer to the Pearson correlation coefficient and the number of
locations used in the validation respectively. The dashed lines are ±15%τ±0.10

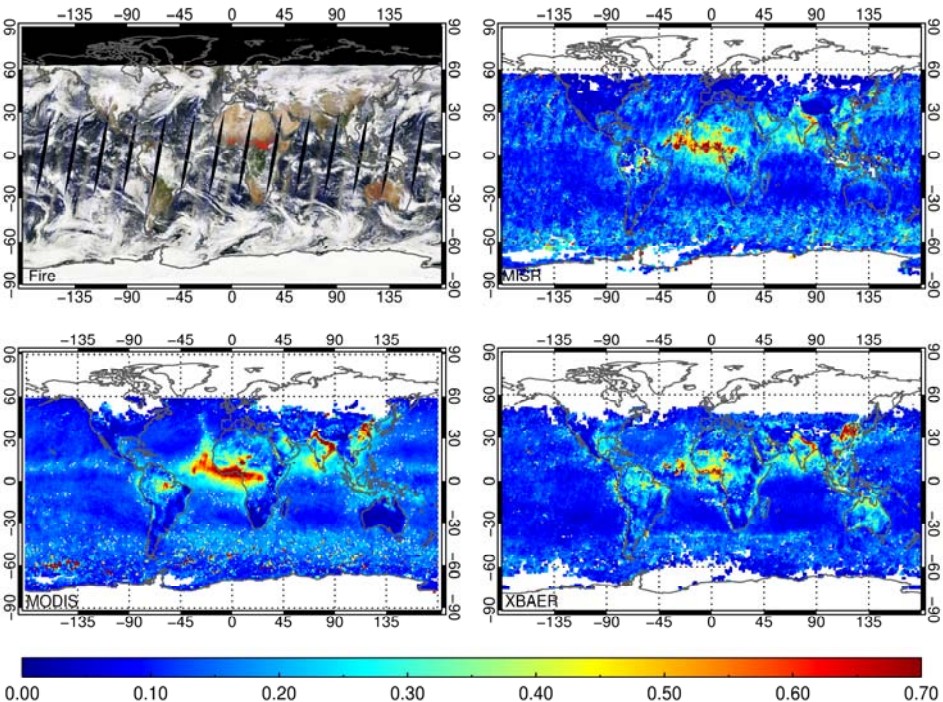

Fig. 4 Comparison of the retrieved global monthly mean AOT at 0.55 μm for December 2016. Upper row: left – MODIS fire product, right-MISR. Lower row: left-MODIS (Dark-Target and DeepBlue combined), right- OLCI (XBAER)

## 4.3 Beijing Haze event observed by OLCI

In the following we show the ability of the retrievals of XBAER used with OLCI data to resolve spatial aerosol patterns on a synoptic scale. A prolonged haze event was observed over Beijing during the period of 16 – 21 December 2016. The intention of applying XBAER to this event is to show the potential of the retrieval to resolve aerosol patterns at a local level and thus being able to support future studies analyzing such events. This event is investigated by both ground-based measurements and satellite observations. Fig.5 (a) shows that winds at the surface were weak with a daily averaged wind speed lower than 3.5 m/s during the period, causing the accumulation of pollutants on a regional scale. The low temperature and high surface pressure near the surface

indicate relatively stable atmospheric conditions in the vertical direction (Fig. 5 (b) (c)). The
dispersion of pollutants out of the boundary layer is therefore slow. The relative humidity
remained high (Fig. 5 (b)) causing the aerosol particles size to increase by the uptake of water
(*Winkler*, 1988), thus making the haze event stronger. Under these meteorological conditions,
pollutants can accumulate over the North China Plain (NCP) (*Li et al.*, 2011).

Fig. 5 (d) shows the time series of concentration of $SO_2$ and $NO_2$ in the boundary layer

provided by ground-based measurements. The concentrations of $SO_2$ and $NO_2$ for haze periods are
three to five times larger than those on relatively clear days. We thus assume anthropogenic
activities to be the major source of AOT. Fig. 5 (e) and (f) show the AOT from AERONET sites
and the time series of the daily mean concentration of $PM_{2.5}$ in Beijing with clearly increased
values in the same timespan (16 – 21 December). The lack of larger AOT values observed by
AERONET is likely going back to too strict cloud screening procedures. The daily mean
concentrations of $PM_{2.5}$ during 16 – 21 December 2016 ranged from 107.1 $\mu g/m^3$ to 394.5 $\mu g/m^3$,
which is far above the daily $PM_{2.5}$ limit of the new threshold value set as the Chinese Ambient Air
Quality                    standard                (75                    $\mu g/m^3$)
(http://transportpolicy.net/index.php?title=China:_Air_Quality_Standards). A large Angstrom
coefficient (Fig. 5(e)) shows that fine particles dominate during this period. In summary, we find
that the cause of the haze event in Beijing and northeastern China goes back to: (1) The stable
meteorological conditions (low wind sppeds and temperature inversion) (2) Local emissions (3)
High relative humidity.

Fig 6 shows the MODIS/Terra derived AOT for the haze period. According to Fig. 6, this

intense part of the haze episode has been partly observed by MODIS. However, a large part of it
(under cloud free conditions) during the first three days is missing, mainly due to cloud masking
applied in the MODIS aerosol retrieval. Fig. 7 shows the AOT from Modern-Era Retrospective
analysis for Research and Applications, Version 2 (MERRA-2) simulation (Rienecker et al., 2011)
in order to exclude the impact from cloud screening. According to Fig. 7, the shape of the area
covered by high AOT in MERRA remains stable except for 20 December, indicating the relative
stable meteorological condition during the haze period. Due to the narrower swath width of OLCI
compared to that of MODIS (1270 km vs 2300 km), OLCI has a longer 'revisit' time for a
repetitive observation of the ground scene in Beijing. According to Fig. 6 and Fig. 8, XBAER
discards fewer clear sky ground scenes than the MODIS retrieval, in particular on the 18th and
19th of December over Eastern China. For this period, the aerosol over Eastern China has been
characterized as 'moderately absorbing aerosol'. The SSA at 0.675 μm from AERONET has
values between 0.88 and 0.91, indicating relatively strong absorption from anthropogenic
activities. The magnitude of AOT for the overlap regions between OLCI and MERRIA are
comparable according to Fig. 7 and Fig. 8. The regions of high aerosol agree well with the aeras
having high $NO_2$ columns in Global Ozone Monitoring Experiment 2 (GOME2) (Richter et al.,
2011) for the corresponding time period as presented in Fig. 9. Fig. 8 illustrates that cloud
masking, surface treatment and aerosol type selection in XBAER all work well for the detection of
extreme haze events. Studies like the one by Zheng et al. (2015), are usually focusing on the origin
of such plumes and the speciation of aerosol particles on a city level. XBAER results utilizing
multi-spectral imagery such as provided by OLCI can support this kind of studies to identify
plume transport and extension. This implies that the OLCI instrument can provide important data
on AOT for atmospheric research.

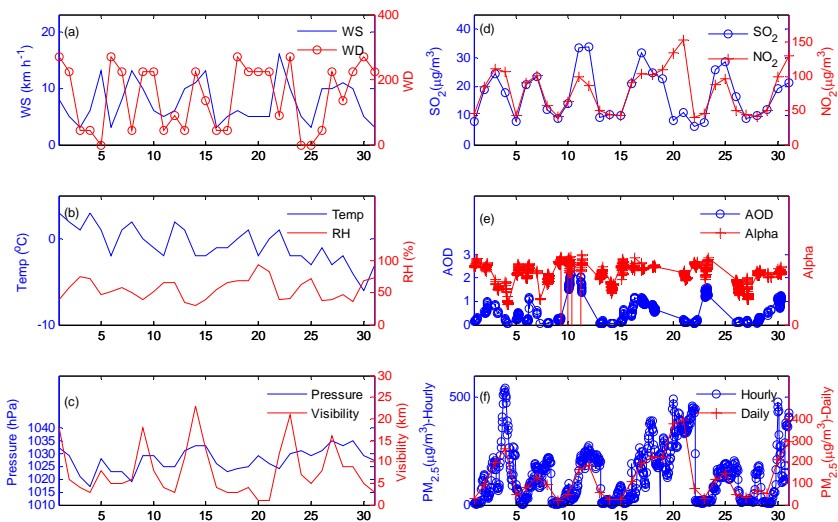


Fig. 5 Time series of meteorological parameters and pollutants during December 2016. (a)wind direction and wind speed (km/h), (b) temperature (˚ C) and relative humidity (%) , (c) atmospheric pressure (hPa) and visibility (m), (d) $SO_2$ and $NO_2$ concentration (μg/m$^3$) (e) AOT and Angstrom coefficient (440-870nm) (Alpha)(f) PM$_{2.5}$ hourly and daily concentration(μg/m$^3$). The atmospheric components and meteorological data are from https://www.aqistudy.cn/historydata/index.php and https://www.wunderground.com/

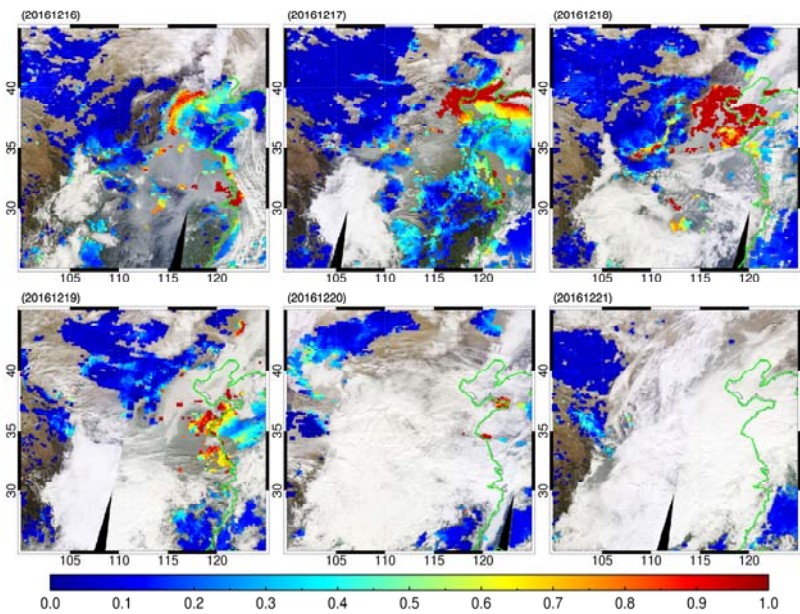

 Fig. 6 Daily MODIS RGB and AOT for East China [100º - 125 º E, 25 º- 45 º N] during 16 – 21

 December 2016 (from top left to bottom right)

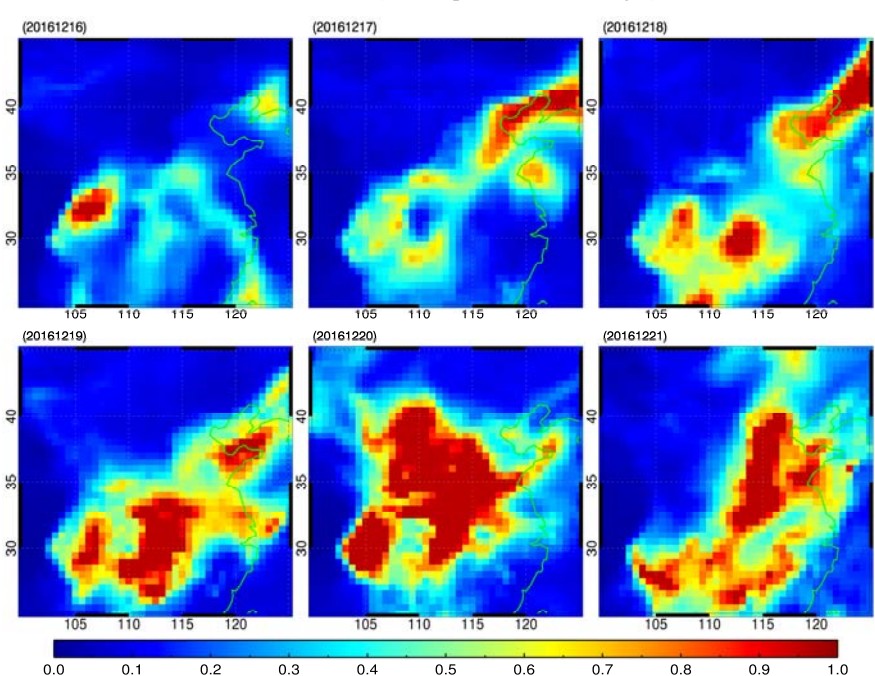

Fig. 7 Same as Fig. 6 but for MERRA AOT


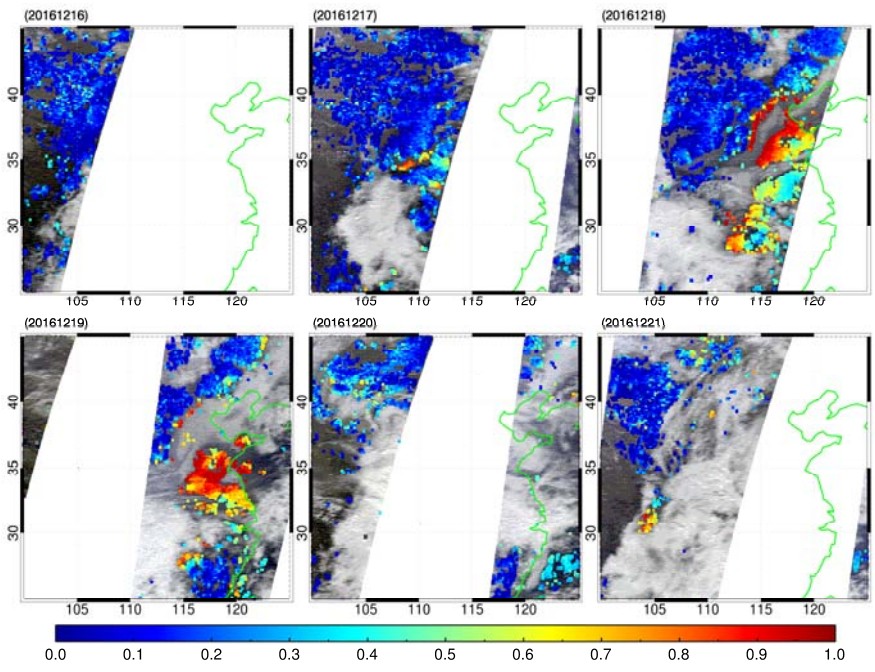


Fig. 8 Same as Fig. 6 but for OLCI



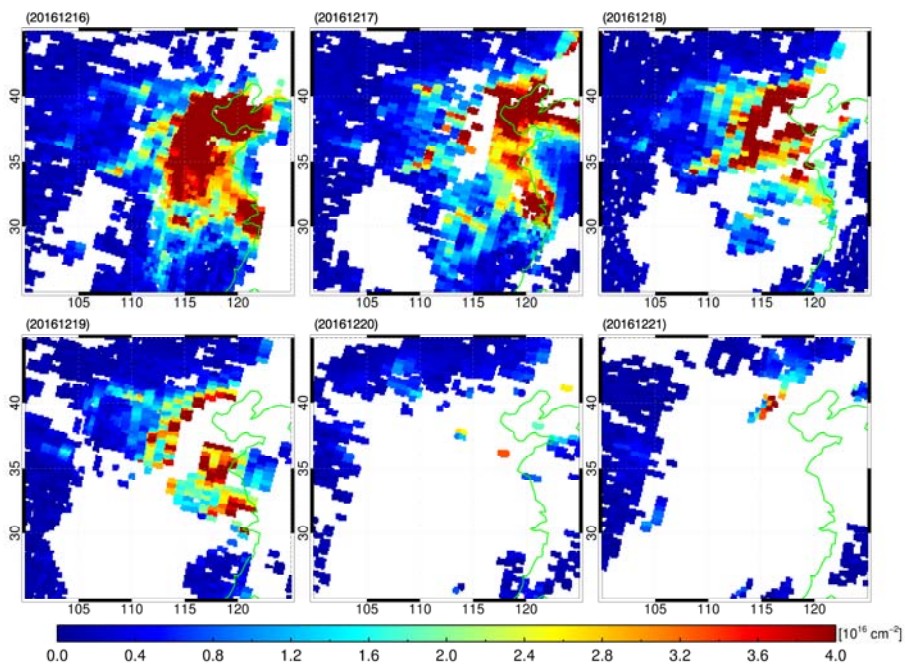


Fig. 9 Same as Fig. 6 but for NO₂ from GOME2a/GOME2b combined IUP-UB product


**5 Discussions**
In this study, we have applied XBAER to data from the OLCI instrument onboard Sentinel-3 for
the first time on both synoptic and global scale. The potential differences caused by different
spectral response functions for OLCI and MERIS have been investigated by using SCIATRAN to
generate representative simulated scenarios for dust aerosol type over desert, moderately
absorbing aerosol over vegetation regions and maritime aerosol over water. The overall
differences for all selected channels for XBAER are smaller than 1.5%. This implies that XBAER
can be used to retrieve AOT from OLCI. Although relatively large differences caused by SRFs
(approximately 20%) have been found for the $O_2A$ channels, the global retrieval of OLCI shows
that the original MERIS cloud masking, which includes the use of $O_2A$ channels, works well for
OLCI and can potentially even be improved as only MERIS-heritage channels have been used so
far with OLCI.
The global monthly mean XBAER AOT maps for December 2016 show good agreement
with those by MODIS and MISR. The comparison with AERONET measurements reveals that
XBAER can provide promising results over both dark and bright surface. The first comparison
with AERONET shows acceptable agreement between the two data sets, with a regression
yielding y = (0.81x ± 0.04) + (0.11 ± 0.01) and correlation of R=0.82. The global retrievals
confirm that XBAER is valid for both dark and bright surfaces because of its use of an optimized
monthly global SSR spectral coefficients dataset.
A significant haze event during December 2016 over Beijing has been analyzed in this paper
based on ground-based and satellite observations to show the potential of the retrieval to resolve
aerosol patterns at a local level and thus being able to support future studies analyzing such events.
This large haze event has been attributed to the large local emissions under unfavorable
meteorological conditions (temperature inversion in vertical direction and no advection). The
MODIS/Terra and OLCI derived AOT both detect the haze event. However, due to cloud
screening, the MODIS AOT partly misses it while OLCI AOT is able to detect the main pattern of
haze for clear conditions. The overlap retrieval for both MODIS and OLCI has similar values,
indicating that OLCI provides another useful data source for air pollution monitoring.
Although the study shows that XBAER can be applied to OLCI observations for synoptic to
global applications, several important issues need to be addressed in the future work. Potential
cloud contamination due to both the relative large calibration uncertainty of OLCI compared to
MERIS as well as the impact of SRF on $O_2A$ channel need to be investigated with the new version
of level 1 TOA reflectance dataset. Modification or improvement for OLCI cloud screening will
be included besides the criteria of brightness, texturing/variability and cloud altitude of the scenes
(Mei et al., 2016b). The underestimation of AOT over regions like Sahara could be explained by
the spheroid dust model adapted from MODIS-DT algorithm due to the impact of non-sphericity
of dust particles on the aerosol phase function (Mei et al., 2016a), a new spheroid model
accounting for aerosol particle non-sphericity will be included in the new version (Dubovik et al.,
2006). The cloud screening evaluation shows that approximately 5 – 10 % clouds may be
misclassified as retrievable clear cases for MERIS (Mei et al., 2016b) which introduces both bias
and potential patchiness of XBAER derived AOT for OLCI. Thus a new cloud post-processing,
following the AATSR dual-view (ADV) algorithm (Sogacheva et al., 2017), will be applied to
discard the pixels that might potentially be affected by cloud (cloud edge, very thin cloud and so
on).


**Acknowledgements**
The authors would like to express their appreciation to Mr. Andreas Heckel from Swansea
University, Dr. Bahjat Alhammoud / Dr. Manuel Arias from ARGANS Company and Dr. Debbie
Richards from EUMETSAT for very valuable and detailed discussion about the OLCI instrument.
The discussion of model simulations from Dr. Anne Blechschmidt and Dr. Abram Sanders from
University of Bremen is highly appreciated. We would also like to express our gratitude to the
AERONET PIs for establishing and maintaining the long-term AERONET stations used for the
validation. The atmospheric components and meteorological data are from
https://www.aqistudy.cn/historydata/index.php and https://www.wunderground.com/. The MODIS
fire point product is available from https://worldview.earthdata.nasa.gov/. We would also like to
thank the anonymous reviewers for their valuable comments, which greatly improved the quality
of this manuscript. The project is partly funded by the University and State of Bremen and the
German Science Foundation (DFG) Trans Regio SFB "Arctic Amplification TR 172". This work
was partly supported by the European Space Agency as part of the Aerosol_CCI project. This
research is in part a contribution by IUP/UB to MARUM a DFG-Research Center/Cluster of
Excellence "The Ocean in the Earth System" (OC-CCP1).

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
