# Peer review of "XBAER derived aerosol optical thickness from OLCI/Sentinel-3 observation"

_Atmospheric Chemistry and Physics, 2017_

## Referee Comment (RC1) · Anonymous Referee #1 · 19 Jun 2017

This manuscript introduces the XBAER method and analysis the AOT result for OLCI/Sentinel-3 using this method. Generally speaking, this manuscript will be a good one after some minor revisions. 1. In the introduction part, there is too much description about the haze itself. Haze is not the main topic of this article, so I suggest the authors to reconsider the content for the haze. 2. Check if all the abbreviations are descripted at the first time, such as "ENVISAT" in Line 142 and "ESA" in Line 145. 3. Only one month for the AOT validation is not enough. Besides, it is lack of large AOT values for validation (only one sample great than 1.2) from the Fig. 3. 4. Line 311: It is better to descript in details which surface and aerosol types is contains in the validation work. 5. Fig.4: Not quite clear to see the fire points in the MODIS fire product.

Sincerely

---

## Referee Comment (RC2) · Anonymous Referee #3 · 11 Sep 2017

General comments:

This paper presents an expansion of XBAER (eXtensible Bremen Aerosol Retrieval) algorithm, which was developed based on previous MERIS (Medium Resolution Imaging Spectrometer), to a new OLCI (Ocean Land Color Instrument) sensor onboard sentinel-3. This contains the details of algorithm and results during December 2016 with specific heavy haze case analysis in Beijing and North China plain region. The scope is well-addressed and the contents are well-organized, thus I recommend it for publication after the responses for some points listed hereafter.

Specific comments:

1) The XBAER algorithm is based on previous works in Mei et al. (2016a; 2016b),

which were applied to MERIS measurement. The authors described the brief explanation of XBAER algorithm in section 3, but some information should be clarified. As the reviewer's understanding, the TOA reflectance, not radiance, is used for the variability test of cloud masking algorithm (line 214) according to the Mei et al. (2016a). And please clarify what is different between cloud height and cloud altitude in line 214. In line 220, please clarify the scale of "space-time" dependent: is the seasonal time scale or day-to-day time scale?

2) Please clarify which channels are used for cloud masking, surface determination, and aerosol inversions as in Mei et al. (2016a). Are the selected channels of OLCI identical with that of MERIS for aerosol retrieval using XBAER although there are additional channels in OLCI?

3) In section 4.1 description of aerosol scenarios, the radius and variance of fine and coarse mode are presented but the fine-mode fraction and refractive indices (or SSA) are not presented. Please clarify whole aerosol microphysical properties for better understanding. Also, which aerosol model is assumed in retrieved OLCI AOT in December over Beijing and NCP region?

4) In line 319-320, the OLCI AOT shows higher R (0.82) compared to the MERIS (0.78), but the MERIS results are in 2009 July according to Mei et al. (2016a). The validation period is different. The period of MERIS and OLCI are not overlapped unfortunately as the authors mentioned in introduction. Then, please compare the accuracy of AOT from OLCI and MERIS for December if possible.

5) Please notify the spatial resolution of the figure 4. Also, the fire product is hard to identify thus larger symbol is required. And, the reviewer checked the MODIS AOT data in figure 4, and it seems to the "AOD_550_Dark_Target_Deep_Blue_Combined_Mean_Mean" product, not Dark-Target-only AOT. Then the MODIS product should be referred as like "Dark-Target (DT) and Deep-Blue (DB) combined product".

6) In line 350, please clarify the "first term". In this paper, a SAVI or NDPI equation is substituted by referred previous studies without detailed explanation. Also, the reason for the DT like over dark surface and DB like over bright surface is insufficient to understand.

7) In section 4.3, please clarify the ground-based measurements: is the one-site data or averaged value of several site within interested domain? Is the UTC or local time in x-axis of figure 5?

Technical correction:

In table 1, please notify the meaning of red colored values in the title of figure.

Line 72: please correct "inlustrate" to "illustrate"

Line 99: please correct "therir" to "their"

Line 121: I think "low wind speeds" is better than just "low winds"

Line 457: please write full name of "SSR", which is not mentioned before.

Also, please check the no space between words (e.g. (Alpha)(f) in line 421) in whole paper.

---

## Author Comment (AC1) · 7 Nov 2017

Dear Editor, dear reviewers,

Thank you for the valuable comments, which have helped us to improve the quality of the paper. The detailed replies to your questions are given below point by point.

Best regards,

Linlu Mei on behalf of all authors

This manuscript introduces the XBAER method and analysis the AOT result for OLCI/Sentinel-3 using this method. Generally speaking, this manuscript will be a good one after some minor revisions.
Response: Thank you for the positive comments.

1. In the introduction part, there is too much description about the haze itself. Haze is not the main topic of this article, so I suggest the authors to reconsider the content for the haze.
Response: The third paragraph in submitted version, describing the important factors which create the severe haze episodes, has been deleted, in response to your criticism.

2. Check if all the abbreviations are described at the first time, such as "ENVISAT" in Line 142 and "ESA" in Line 145.
Response: All abbreviations have been thoroughly checked.

3. Only one month for the AOT validation is not enough. Besides, it is lack of large AOT values for validation (only one sample great than 1.2) from the Fig. 3.
Response: This study uses the first month of data released and the results are very promising. The comparison with the AOT data products from AERONET, MODIS and MISR show good agreement (e.g. Pearson correlation coefficient 0.82). The focus of this study is to describe the retrieval algorithm and the results to demonstrate the feasibility of using XBAER on invert the reflectances measured at the top of the atmosphere by the new sensor- OLCI on sentinel 3. As we mentioned in the paper, potential cloud contamination due to both the relatively large calibration uncertainty of OLCI (can reach 6%) compared to MERIS as well as the impact of SRF on $O_2A$ channel need to be investigated with the new version of level 1 TOA reflectance dataset after (not yet) released. An additional validation paper and corresponding applications papers will be prepared after the new run with the new calibrated level 1 TOA dataset. In order to illustrate the monthly variability of XBAER-derived AOT, the following additional two figures which show monthly mean AOT of Jan., Feb. of 2017 are presented. We can see that XBAER-derived AOTs are quite similar to other operational AOT products (MODIS_DarkTarget_DeepBlue_Combined, MISR and MODIS_DeepBlue_Land) for the winter season of 2016 (Dec 2016 – Feb. 2017).

[Figure]

[Figure]

Fig. 1 Monthly mean AOT of (a) MODIS_DarkTarget_DeepBlue_Combined (upper left) (b) MISR (upper right) (c) MODIS_DeepBlue_Land (lower left) (d) XBAER (lower right) for Jan. and Feb. 2017.

4. Line 311: It is better to descript in details which surface and aerosol types is contains in the validation work.

Response: The collocations of Fig.3 (in the paper) contain various surface and aerosol types, which ensure a wide representativeness of the validation. Fig 2 below shows global distribution of collocated AERONET observations with OLCI and corresponding surface types for December 2016. According to Fig.2, the validation of Fig 3 (in the paper) represents all major surface types worldwide. Fig. 3 show the aerosol types used in XBAER retrieval, according to the AERONET sites distributions in Fig.2, we can see that the validation figure also contains all proposed aerosol types, they are weakly absorbing, moderately absorbing, strongly absorbing and dust. Some more details have been included in the revised paper.

[Figure]

Fig 2. Global distribution of collocated AERONET observations with OLCI and corresponding surface types for December 2016

[Figure]

Fig.3 Aerosol types over land used in the XBAER algorithm designated at 1°×1° grid for different seasons.

The four sub-figures represent four seasons. Upper row: left – December, January and February (DJF), right-March, April and May (MAM). Lower row: left-June, July and August (JJA), right- September, October and November (SON).

5.   Fig.4: Not quite clear to see the fire points in the MODIS fire product.

Response: Fig. 4 has been updated.

---

## Author Comment (AC2) · 7 Nov 2017

Dear Editor, dear reviewers,

Thank you for the valuable comments, which have helped us to improve the quality of the paper. The detailed replies to your questions are given below point by point.

Best regards,

Linlu Mei on behalf of all authors

General comments:

This paper presents an expansion of XBAER (eXtensible Bremen Aerosol Retrieval) algorithm, which was developed based on previous MERIS (Medium Resolution Imaging Spectrometer), to a new OLCI (Ocean Land Color Instrument) sensor onboard sentinel-3. This contains the details of algorithm and results during December 2016 with specific heavy haze case analysis in Beijing and North China plain region. The scope is well-addressed and the contents are well-organized, thus I recommend it for publication after the responses for some points listed hereafter.

Specific comments:

The XBAER algorithm is based on previous works in Mei et al. (2016a; 2016b), which were applied to MERIS measurement. The authors described the brief explanation of XBAER algorithm in section 3, but some information should be clarified. As the reviewer's understanding, the TOA reflectance, not radiance, is used for the variability test of cloud masking algorithm (line 214) according to the Mei et al. (2016a). And please clarify what is different between cloud height and cloud altitude in line 214. In line 220, please clarify the scale of "space-time" dependent: is the seasonal time scale or day-to-day time scale?

Response: Thanks for the comments. We have changed "radiance" to "reflectance". "height" and "altitude" have the same meaning here and we keep "height" in the revised version. The spatial and temporal resolutions of the database are 10 km and monthly, respectively.

1) Please clarify which channels are used for cloud masking, surface determination, and aerosol inversions as in Mei et al. (2016a). Are the selected channels of OLCI identical with that of MERIS for aerosol retrieval using XBAER although there are additional channels in OLCI?

Response: We have used the same "overlapped" channels between OLCI and MERIS as we described in Line 244 in original version. One more column has been included as presented in Table 1.

2) In section 4.1 description of aerosol scenarios, the radius and variance of fine and coarse mode are presented but the fine-mode fraction and refractive indices (or SSA) are not presented. Please clarify whole aerosol microphysical properties for better understanding. Also, which aerosol model is assumed in retrieved OLCI AOT in December over Beijing and NCP region?

Response: The fine/coarse mode volumes ($\mu m^3/\mu m^3$), which was used to calculate the fine mode fraction, are 0.055/0.038, 0.056/0.057, 0.051/0.040, 0.02/0.157 for weakly absorbing, moderately absorbing, strongly absorbing and dust, respectively. The corresponding SSAs are 0.92, 0.95, 0.87 and 0.95. Aerosol types over land used in the XBAER algorithm designated at 1°×1° grid for different seasons are presented in Fig. 3 below. According to Fig.3, moderately absorbing and weakly absorbing were used for the NCP region for December. All above details have been included in the revised version.

[Figure]

Fig.3 Aerosol types over land used in the XBAER algorithm designated at 1°×1° grid for different seasons. The four sub-figures represent four seasons. Upper row: left – December, January and February (DJF), right-March, April and May (MAM). Lower row: left-June, July and August (JJA), right- September, October and November (SON).

3)  In line 319-320, the OLCI AOT shows higher R (0.82) compared to the MERIS (0.78), but the MERIS results are in 2009 July according to Mei et al. (2016a). The validation period is different. The period of MERIS and OLCI are not overlapped unfortunately as the authors mentioned in introduction. Then, please compare the accuracy of AOT from OLCI and MERIS for December if possible.

Response: The correlations coefficient with AERONET is about 0.70 for December, 2009 for XBAER version 1.6 product, we are still ongoing for the reprocessing of XBAER version 2.0.

4)  Please notify the spatial resolution of the figure 4. Also, the fire product is hard to identify thus larger symbol is required. And, the reviewer checked the MODIS AOT data in figure 4, and it seems to the "AOD_550_Dark_Target_Deep_Blue_Combined_Mean_Mean" product, not DarkTarget-only AOT. Then the MODIS product should be referred as like "Dark-Target (DT) and Deep-Blue (DB) combined product".

Response: The figure and tile have been improved.

5)  In line 350, please clarify the "first term". In this paper, a SAVI or NDPI equation is substituted by referred previous studies without detailed explanation. Also, the reason for the DT like over dark surface and DB like over bright surface is insufficient to understand.

Response: The definition of SAVI and NDPI are given below:

$$SAVI = \frac{R(\lambda_{14}) - R(\lambda_7)}{R(\lambda_{14}) + R(\lambda_7) + L}(1 + L),$$ (1)

$$L = 1 - \frac{2R(\lambda_{14}) + 1 - \sqrt{(2R(\lambda_{14}) + 1)^2 - 8(R(\lambda_{14}) - R(\lambda_7))}}{2},$$ (2)

where R is the SSR and the subscript for the wavelength denotes the MERIS channel numbers defined in Table 1.

$$NDPI = \frac{R(\lambda_2) - R(\lambda_5)}{R(\lambda_3)},$$ (3)

XBAER uses a one-parametric approximation of SSR, it is assumed that the SSR for a given surface type at a given wavelength is determined by the use of a linear relationship to a particular and selected vegetation index, SAVI, if SAVI values are very small over regions like desert, the non-vegetation component (B vector as presented in Mei et al., 2016b) dominates the surface treatment, and XBAER becomes a DeepBlue-similar aerosol retrieval algorithm. If SAVI values are large over regions like Amazon, the vegetation component (A vector as presented in Mei et al., 2016b) dominates the surface treatment, and XBAER becomes a DarkTarget-similar aerosol retrieval algorithm.